# Experimental Validation of the Numerical Model for Oil–Gas Separation

**Sorin Gabriel Tomescu [1,2], Ion Mălăel [1,*], Rareș Conțiu [1] and Sebastian Voicu [1]**

1   National Research and Development Institute for Gas Turbines COMOTI, 220 D Iuliu Maniu Bd., Sector 6, 061126 Bucharest, Romania; sorin.tomescu@comoti.ro (S.G.T.); rares.contiu@comoti.ro (R.C.); sebastian.voicu@comoti.ro (S.V.)
2   Power Engineering Faculty, University Politehnica of Bucharest, Splaiul Independentei No. 313, Sector 6, 060042 Bucharest, Romania
*   Correspondence: ion.malael@comoti.ro; Tel.: +40-21-434-02-40

**Abstract:** The oil and gas sector is important to the global economy because it covers the exploration, production, processing, transportation, and distribution of oil and natural gas resources. Despite constant innovation and development of technologies to improve efficiency, reduce environmental impact, and optimize operations in the gas and oil industry over the last few decades, there is still room to increase the efficiency of the industry's equipment in order to reduce its carbon footprint. The separation of gas from oil is a critical stage in the technological production chain, and it is carried out using high-performance multi-phase separators to limit greenhouse gas emissions and have a low impact on the environment. In this study, an improved gas–oil separator configuration was established utilizing CFD techniques. Two separator geometry characteristics were studied. Both cases have the same number of subdomains, two porous media, and four fluid zones, but with a difference in the pitch of the cyclone from the inlet subdomain. The streamlines in a cross-plan of the separator and the distribution of the oil volume fraction from the intake to the outlet were two of the numerical results that were shown as numeric outcomes. The validation of these results was performed using an experimental testing campaign that had the purpose of determining the amount of lubricating oil that is discharged together with the compressed gas at the separator outlet.

**Keywords:** gas–oil separator; CFD; oil volume fraction; streamlines; porous media; experimental campaign





## 1. Introduction

The exploration, production, refining, and distribution of petroleum and natural gas resources are all part of the global oil and gas industry, which is essential to the world economy because it provides energy for a variety of uses, including heating, electrical production, and transportation [1]. It is vital to note that the oil and gas industry is evolving due to mounting concerns about climate change and the global movement toward renewable energy sources. The industry is looking into innovative technologies [2] with a smaller influence on the climate change process in order to adapt to changing market dynamics and minimize its environmental impact [3].

In the petroleum sector, a process called oil and gas separation involves separating natural gas and crude oil from one another and other components of the combination [4]. This procedure is essential for turning natural gas and crude oil into useful goods. The main objective is to remove water, sediments, and other contaminants in order to concentrate the valuable oil and gas components [5].

Depending on the facility's unique requirements and the mixture's composition, several techniques and pieces of equipment are used to separate oil from gas [6]. Gravity separation or mechanical separation are examples of typical approaches [7,8].While mechanical separation techniques use physical barriers or devices to separate oil and gas from

the mixture, gravity separators, such as settling tanks or vessels, use the density difference between oil, gas, and other components to separate them [9].

Oil and gas industry equipment called two-phase oil–gas separators are used to separate oil and gas from a mixed stream of fluids [10]. They are often utilized in upstream activities where extracted fluids comprise a combination of gas and oil, such as oil-producing fields or gas processing facilities [11].

A two-phase separator's, presented in Figure 1, primary function is to effectively separate the oil and gas phases so that they can be treated or transported separately [12]. The variations in density and volatility between oil and gas are beneficial for the separation process. Through an entrance nozzle, the mixed oil–gas stream is introduced to the separator. To generate a cyclonic effect, the flow may be directed vertically or tangentially, which helps with the separation process [13]. The flow velocity drops inside the separator, enabling gravity to take over. The difference in density causes the oil and gas to begin to separate. Being lighter than the oil phase, the gas phase has a tendency to climb to the top [14].

To keep the ideal working conditions, level control devices are frequently used in two-phase separators. The separator's liquid level is controlled by these controls to guarantee effective separation and stop liquid from leaking into the gas outlet [15]. Factors including flow rates, fluid characteristics, separator shape, and operating parameters affect the separator's efficiency [16].

Separators commonly use additional components and methods, such as gravity settling, baffles, and demisting devices, to increase the overall efficacy and efficiency of the separation process. A porous medium may significantly improve the separation process [17]. Porous media is commonly usedin two major forms in oil and gas separators: filtering media and coalescing media.

Solid particles are captured by a filtering medium, preventing them from passing through the separator. Sand, rust, scale, and other debris that could be in the fluid stream are examples of these particles. Filtering media are made with certain pore diameters that permit gas and liquid to pass through while trapping solid particles [18].

To decrease the accumulation of tiny oil droplets floating in the fluid stream, coalescing media are used. The media's porous structure gives the droplets a lot of surface area to interact with one another and combine to produce bigger droplets. These bigger droplets may be collected more successfully and are simpler to separate from the fluid.

Materials like fiberglass, polypropylene, and metal wire mesh are frequently used to create coalescing media [19,20].

Particle size distribution, desired separation efficiency, operating circumstances, and the properties of the fluid stream are only a few of the variables that affect the choice of porous media [21] and how it is configured in an oil and gas separator.

The mechanical methods used to remove oil from compressed gas in gas–oil separators are gravity separation, separation by changing the direction or speed of flow, separation using centrifugal force, and separation using coalescence. Each of the methods listed requires a particular mounting configuration and has limitations in retaining oil droplets of varying sizes. Gas–oil separators used in industry can be constructed using one or more separation methods.

When designing oil–gas separators, several parameters such as fluid characteristics, flow rates, operating circumstances, and separation needs must be taken into account [22]. The separation requirements, fluid characteristics, flow rates and operating conditions, internal design, pressure drop, and installation and maintenance are all important aspects of the separator design process [23].

CFD (computational fluid dynamics) analysis is widely used in the design and optimization of two-phase separators [24,25]. Numerical simulations provide important insights into the flow behavior and separation effectiveness of two-phase separators by representing the flow patterns inside the separator, including the distribution of gas and oil phases [26]. This allows researchers to identify locations of considerable turbulence

or recirculation that may affect separation efficiency. CFD simulations can determine the separation performance of a two-phase separator by following the paths of gas and oil particles [27]. By evaluating residence time and separation performance, researchers may alter design parameters such as inlet/outlet combinations, baffle designs, and internal geometries to achieve the needed separation efficiency [28].

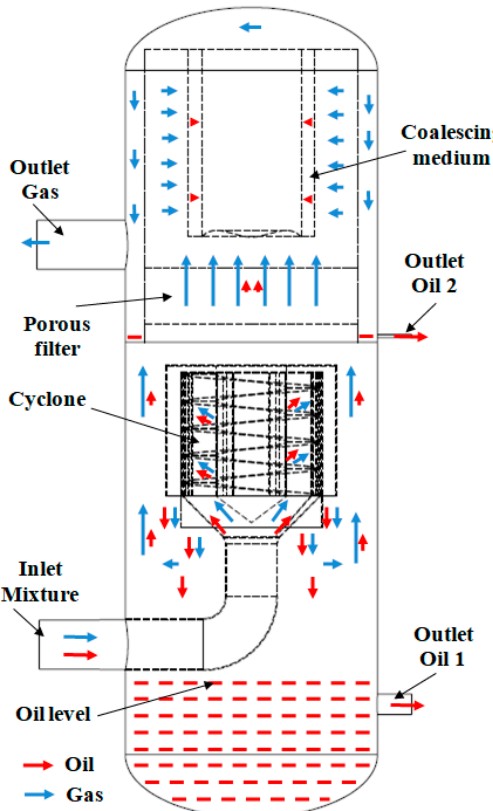

**Figure 1.** Two-phase separator.

Recently, Wang et al. [29] published a paper in which they quantitatively investigated the oil–gas separation in a separator using the Eulerian–Lagrangian approach. Acharya et al. [30] published another academic study in which CFD simulations were used to examine the effectiveness of a horizontal divider. Krzemianowski et al. [31] provided intriguing numerical research on gas–oil–water combination separation.Overall, CFD analysis is a great tool for understanding complicated fluid dynamics and optimizing two-phase separation designs [32]. It helps researchers make more informed judgments, increase separation performance, and cut expenses associated with physical prototyping using trial and error [33].

Experimental tests are carried out to confirm the numerical results and evaluate the performance and efficiency of these separators [34]. The separation efficiency test, which measures a separator's capacity to successfully separate oil and gas, is the most significant test that must be performed during an experimental test campaign [35]. The inlet port is used to inject the working fluid into the separator, and the outputs are evaluated to determine the separation efficiency of each phase.

Machado et al. [36] conducted an interesting experiment in which they investigated the evolution of an oil–gas mixture using a three-separator arrangement. The flow behavior was studied using a radiotracer approach. The gas separation analyses demonstrated that the impulse radiotracer approach detects non-negligible malfunctions but is not sensitive enough to regulate the zero tolerance of oil content.

Wang et al. [37] carried out tests to examine the separation efficiency, pressure loss, and particle size distributions of the droplets at the inlet and outlet of the separator using a

unique cyclone separator with multi-layer central channels. In order to comprehend the separation process and determine the primary factors impacting the separation efficiency, Yang et al. [38] performed an experimental investigation of an oil–gas cyclone separator in an oil-injection compressor system. The performance of the separators was evaluated using a Malvern particle size analyzer. The separation performance was evaluated by monitoring the oil droplet size distribution and oil concentration both upstream and downstream of the separators. The study's findings aided in the creation of an improved cyclone separator.

Another test is the pressure drop test [39], which measures the pressure drop across the separator by monitoring the separator's inlet and output pressures while altering the flow rate. The pressure drop is a key measure for evaluating the separator's performance and determining if it meets the pressure requirements of the system.

The particle size distribution test, although the most difficult experimental test, is required in the efficiency validation procedure of a separator [40]. The inclusion of solid particles, such as sand or scale, can impair separator performance in some instances. Particle size distribution tests measure the size and concentration of solid or oil particles in the working fluid of a separator [41].

These studies are carried out utilizing laboratory-scale or pilot-scale separators that are as similar to real-world working conditions as feasible. The information gleaned from these tests assists engineers and operators in optimizing separator design and operation for efficient oil and gas production.

Therefore, CFD methods are utilized in this work to evaluate the unsteady flow in an oil–gas separator. The major focus of the 3D numerical simulation is determining the oil trajectory withthe complicated geometry of the separator and the behavior of the porous media utilized in the filter domains. The numerical analysis findings are validated using experimental tests performed on a screw compressor testing bench using an oil–gas separator to separate the oil–gas combination exiting the compressor pressure side. The amount of oil is monitored after several hours of operation.

## 2. Materials and Methods

The governing equations for each phase must be solved, and interactions at the phase interfaces must be taken into consideration when simulating multiphase flows using CFD. The Navier–Stokes equations, which explain the conservation of mass, momentum, and energy for each phase, are the key equations at play. Other equations, such as the volume of fluid (VOF) technique, the level set method, and the discrete phase model (DPM), are also used to model the phase interfaces.

The dynamics at the phase boundaries, handling phase transitions, and addressing complicated geometries are all difficult to simulate in multiphase flows, especially for large-scale simulations where the computing cost might be rather high. CFD software, created exclusively for multiphase flow analysis, offers a variety of modeling tools and algorithms intended to meet these issues. CFD simulations may be used by researchers to optimize designs, analyze fluid behavior under diverse situations, and gain insights into the underlying physics of complicated multiphase flow processes. The computational domain of the two geometric features that we numerically and experimentally investigated in this study is shown in Figure 2, and we employed the boundary conditions shown in Figure 3 for the computational fluid dynamic simulations with the working fluid properties presented in Table 1.

The calculation grid, shown in Figure 4, was generated using methods based on the maximum element size of 20 mm, generating calculation grids with 4,261,516 nodes and 11,906,316 elements. For local grid control, a sizing condition was applied, having the minimum element of 3 mm at the interface with the spiral and intermediate separator cylinder and for the surfaces of the demister and separator filter.

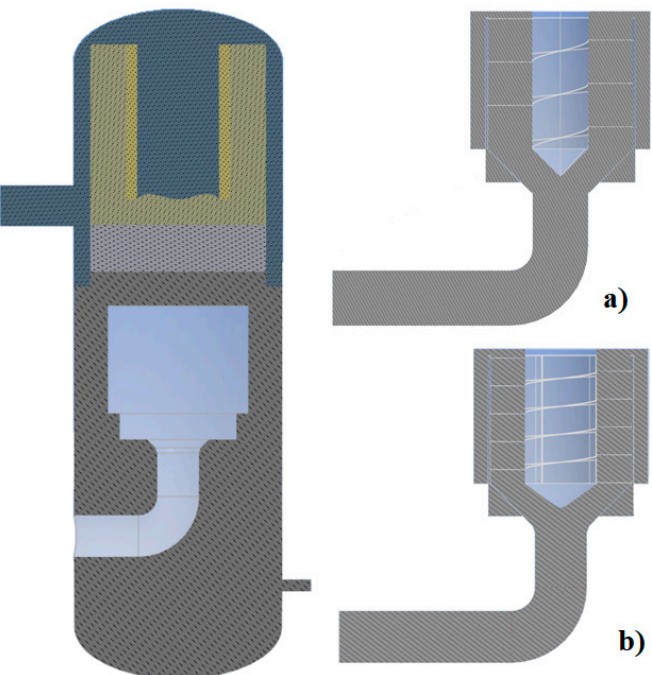

**Figure 2.** Computational domain: (**a**) case 1 cyclone pitch 3 and (**b**) case 2 cyclone pitch 4.5.

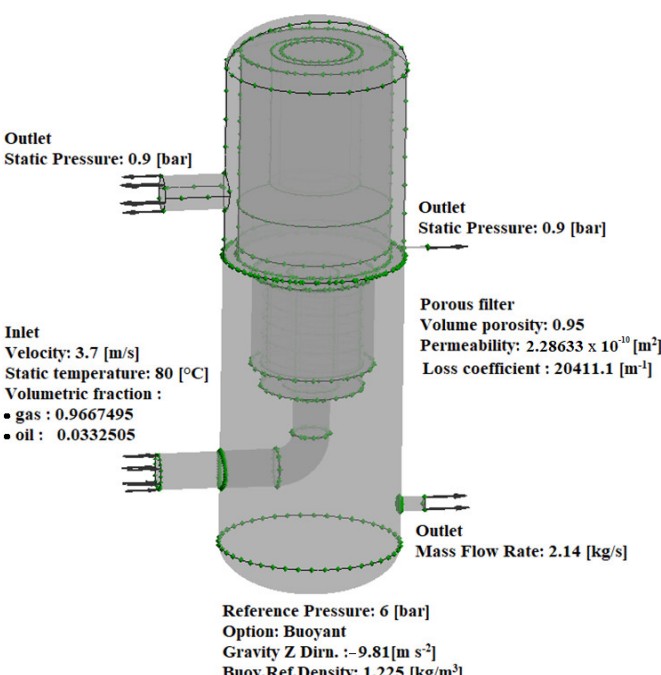

**Figure 3.** Boundary conditions.

The turbulence model used in the unsteady flow simulation was *k*-ε, and the two transport equations are presented below. The transport equation for turbulent kinetic energy (k) is:

$$\frac{\partial k}{\partial t} + \overline{u}\frac{\partial k}{\partial x} + \overline{v}\frac{\partial k}{\partial y} + \overline{w}\frac{\partial k}{\partial z} = \frac{1}{\rho}\left[\frac{\partial}{\partial x}\left(\left(\mu + \frac{\mu_t}{\sigma_k}\right)\frac{\partial k}{\partial x}\right) + \frac{\partial}{\partial y}\left(\left(\mu + \frac{\mu_t}{\sigma_k}\right)\frac{\partial k}{\partial y}\right) + \frac{\partial}{\partial z}\left(\left(\mu + \frac{\mu_t}{\sigma_k}\right)\frac{\partial k}{\partial z}\right)\right]$$
$$+ \frac{1}{\rho}\sum_{i,j}\tau_{tij}\frac{\partial \overline{u_i}}{\partial x_j} - \varepsilon \tag{1}$$

The transport equation for the dissipation rate of turbulent energy (ε) is:

$$\frac{\partial \varepsilon}{\partial t} + \overline{u}\frac{\partial \varepsilon}{\partial x} + \overline{v}\frac{\partial \varepsilon}{\partial y} + \overline{w}\frac{\partial \varepsilon}{\partial z} = \frac{1}{\rho}\left[\frac{\partial}{\partial x}\left(\left(\mu + \frac{\mu_t}{\sigma_\varepsilon}\right)\frac{\partial \varepsilon}{\partial x}\right) + \frac{\partial}{\partial y}\left(\left(\mu + \frac{\mu_t}{\sigma_\varepsilon}\right)\frac{\partial \varepsilon}{\partial y}\right) + \frac{\partial}{\partial z}\left(\left(\mu + \frac{\mu_t}{\sigma_\varepsilon}\right)\frac{\partial \varepsilon}{\partial z}\right)\right]$$
$$+ C_{\varepsilon 1}\frac{\varepsilon}{\rho k}\sum_{i,j}\tau_{tij}\frac{\partial \overline{u_i}}{\partial x_j} - C_{\varepsilon 2}\frac{\varepsilon^2}{k}$$

(2)

where: $\mu_t$ represents the turbulent viscosity coefficient; $C_{\varepsilon 1}$, $C_{\varepsilon 2}$ are empirical constants, and $\sigma_k$ and $\sigma_\varepsilon$ are Prandtl numbers.

Also $C_{\varepsilon 1} = 1.44$ , $C_{\varepsilon 2} = 1.92$ , $\sigma_k = 1.0$ , $\sigma_\varepsilon = 1.3$ .

**Table 1.** Working fluid Properties.

|  | Unit | Gas | Oil |
|---|---|---|---|
| Molar Mass | kg/kmol | 28.96 | 309 |
| Density | kg/m$^3$ | 1.225 | 827.1 |
| Specific heat capacity | J/kg/K | 1004.4 | 2124 |
| Ref. Temperature | K | 298.15 | 353.15 |
| Ref. Pressure | bar | 1 | 1 |
| Ref. Spec. Enthalpy | J/kg | 0 | 156,801 |
| Ref. Spec. Entropy | J/kg/K | 0 | 503 |
| Dynamic Viscosity | kg/m/s·10$^{-3}$ | 18.31 | 11.33 |
| Thermal Conductivity | W/m/K | 0.0261 | 0.1274 |

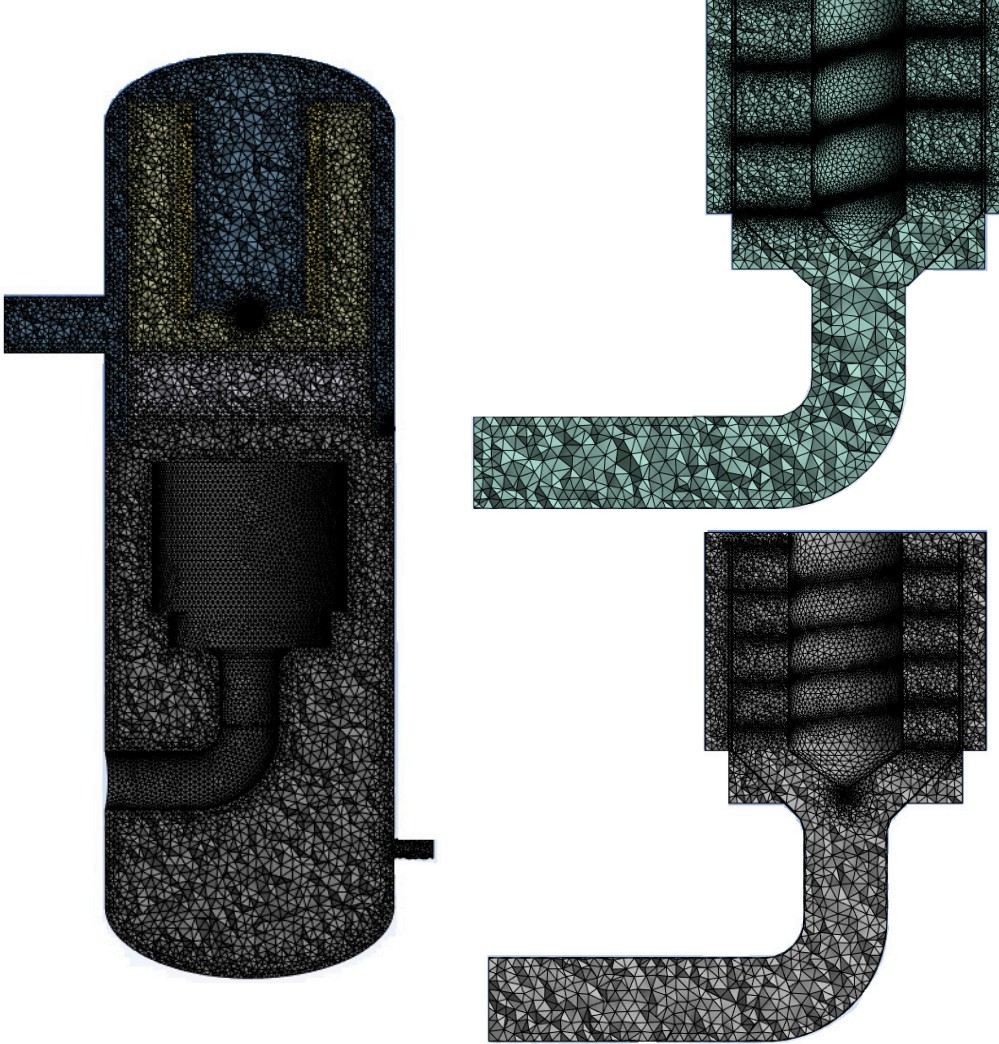

**Figure 4.** Mesh.

Numerical validation of the results was carried out on the test bench of a screw compressor test bench (Figure 5), which provides an excellent setting for conducting research and identifying the functional requirements and performance characteristics of the gas–oil separation solution. This test bench enables the manipulation of key functional parameters within a broad range, allowing for comprehensive investigations.

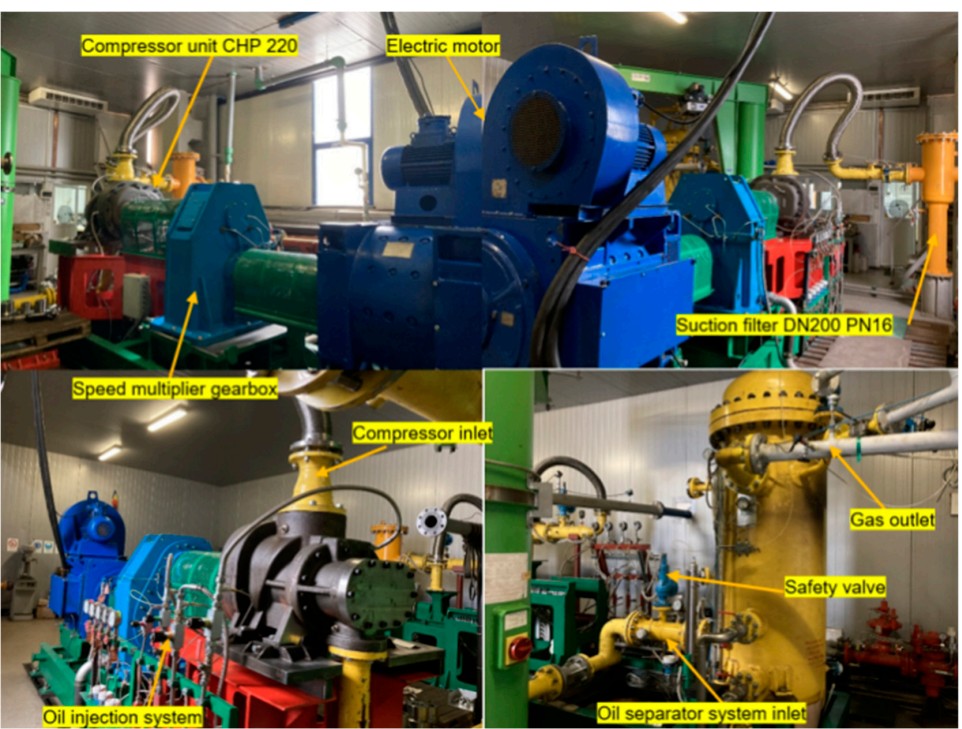

**Figure 5.** Experimental test bench.

The acquisition of functional parameter measurements was facilitated withthe conversion of electrical signals from process transducers mounted on the test rig using a programmable logic controller (PLC—Figure 6). The parameters of interest were recorded at a resolution of one-second intervals and subsequently stored and exported as an Excel file.

To assess the overall effectiveness of the separation system, the suspended droplets present in the compressed air at the separation system's outletwere collected for analysis. The residual oil content in the compressed air was determined using a gravimetric method. The experimental setup included a coalescer filter with a nominal diameter of 400 mm and a nominal pressure of 40 bar.

This coalescer filter comprises two retention stages, namely, an inertial stage and a stage equipped with five 6CU-280 × 1 coalescer filters manufactured by Parker Finite. These filter elements exhibit the capability to capture 100% of droplets larger than 1 μm.

A differential pressure gauge was installed to effectively monitor the coalescing filter's performance, enabling the observation of pressure drop variations across different operation conditions.

Prior to commencing the experimental tests, meticulous preparation was undertaken, involving the thorough cleaning of the coalescer filter housing, alongside the replacement of its filter elements with pristine ones.

To facilitate independent oil collection, the system is equipped with two dedicated connections, each equipped with manual valves, ensuring precise control over the collection process.

During the initial stage of inertial separation, a minimal amount of oil is anticipated to be collected. This is attributed to the small size of oil droplets that elude being captured by centrifugal forces during the first step of the separation process.

During the subsequent separation step, a highly effective retention system comes into play, capturing all droplets with a size exceeding 1 μm, thereby significantly enhancing the separation efficiency of the overall solution.

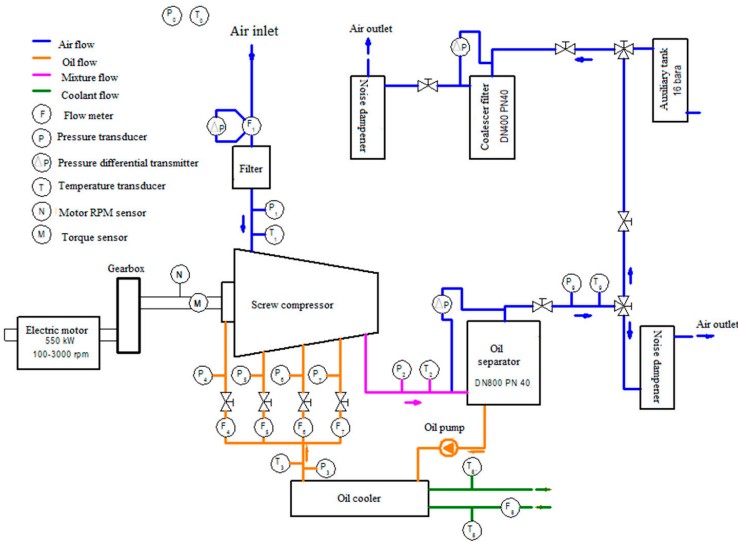

**Figure 6.** The block diagram showing the experimental setup.

The following parameters of interest were meticulously recorded to facilitate future in-depth analysis of the gas–oil separation solution's performance:

1.  Pressure and temperature of the compressed air upon discharge and at the inlet to the separator vessel.
2.  Compressor shaft speed, which plays a vital role in the overall system dynamics.
3.  Pressure and temperature of the compressed air upon exiting the separator vessel, which are critical indicators of the oil separation overall efficiency.
4.  Power consumption by the electric motor that drives the compressor, which offerinsights into energy utilization.
5.  Oil pressure and temperature measurements following the oil cooler, which assessthe oil's thermodynamic characteristics.
6.  Air inlet flow rate, which iscrucial for understanding the overall test bench dynamics and efficiency.
7.  Oil flow rates at specific oil injection points, including seals, inlet bearings, rotors, and discharge bearings, which enable a comprehensive view of the oil distribution throughout the system.
8.  Oil pressure readings at each oil injection point, which providevaluable data on the oil's behavior and its impact on system performance.

## 3. Results

Given that multiphase flow simulations can be computationally intensive and call for a solid grasp of fluid dynamics, numerical methods, and the physics of the simulated process, the accuracy of the CFD simulation results depends on the caliber of the selected model and the input parameters used. Tracking the interface between phases properly is one of the difficulties in multiphase simulation. In our study, the interface tracking was accomplished using the volume of fluid (VOF) method. This technique aids in capturing the morphing and shifting of interfaces.

An approach for visualizing fluid flow patterns in computational fluid dynamics (CFD) simulations is the use of streamlines. They provide a visual representation of the paths that fluid particles might take via a flow field. The direction and course that fluid elements take at various sites in the fluid domain are revealed by streamlines.

The streamlines for both situations are shown in Figure 7 for positions ranging from 0 s to 20 s. The oil–gas separator geometry is complicated; therefore, it has several recirculation zones that may be seen. However, because of the extremely low velocity, these zones have little impact on the separator's effectiveness, with a pressure drop of less than 0.5 bar.

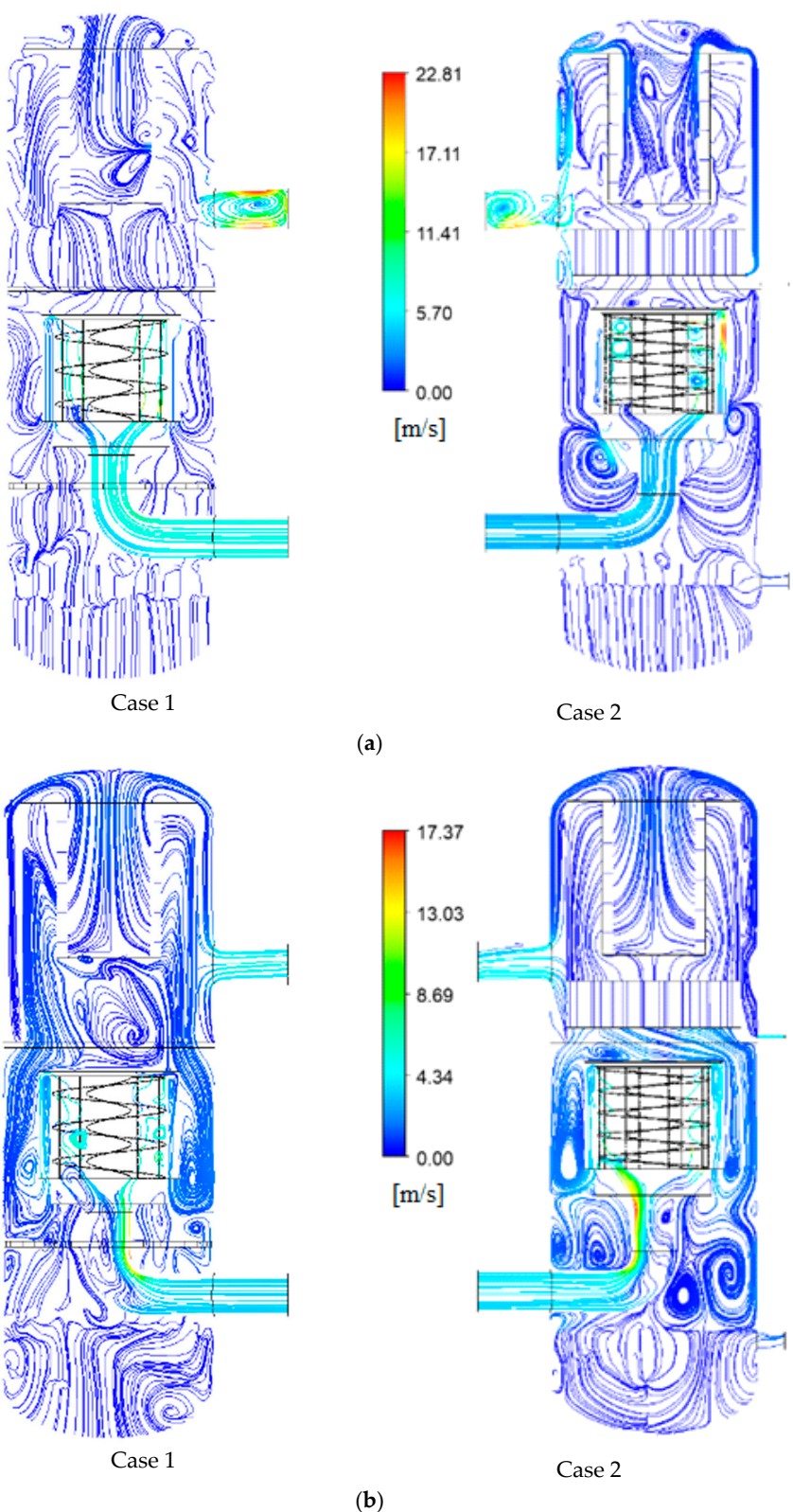

**Figure 7.** *Cont.*

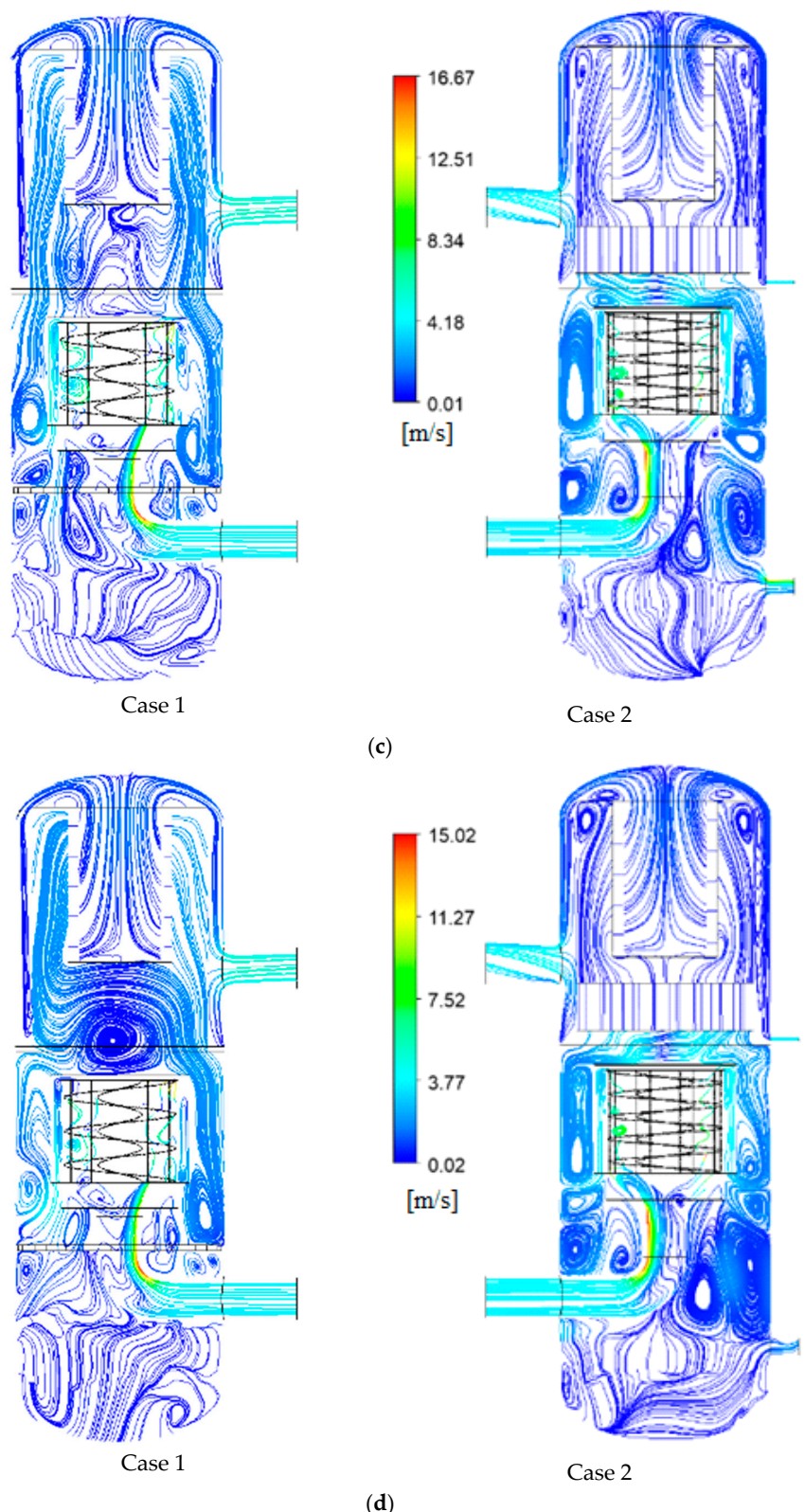

**Figure 7.** *Cont.*

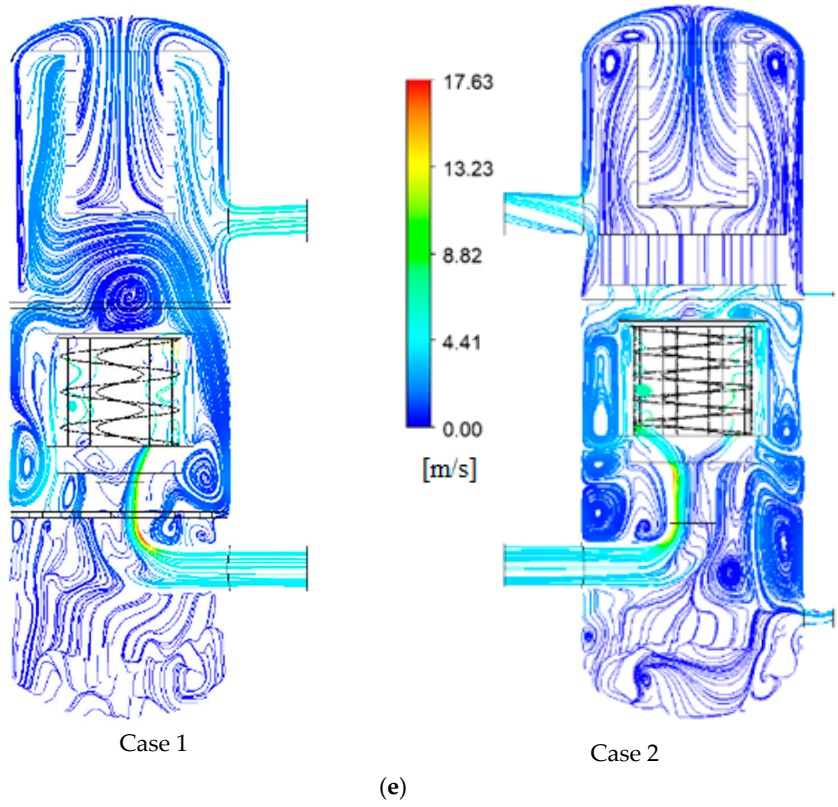

Case 1

Case 2

(**e**)

**Figure 7.** Velocity streamlines: (**a**) 0 s; (**b**) 5 s; (**c**) 10 s; (**d**) 15 s; and (**e**) 20 s.

Due to the dynamic nature of fluid flows that may occur in the oil–gas separator, the oil volume fraction can change over time and space. For this reason, it is crucial to track and analyze how the volume fraction changes in different parts of the domain to comprehend the behavior of the multiphase system. The oil volume fraction for the two investigated examples is shown in Figure 8, and it can be seen that the high pitch of the inlet domain spiral increases separator efficiency by pushing the oil to the separator's bottom through the spiral's lateral gaps. Figure 8e demonstrates that in the second scenario, the oil falls gravitationally through the lateral gaps rather than ascending to the top of the spiral. When there is little oil at the top of the spiral, only a tiny portion of the oil makes it to the outlet after passing through two filters, which are represented by porous media.

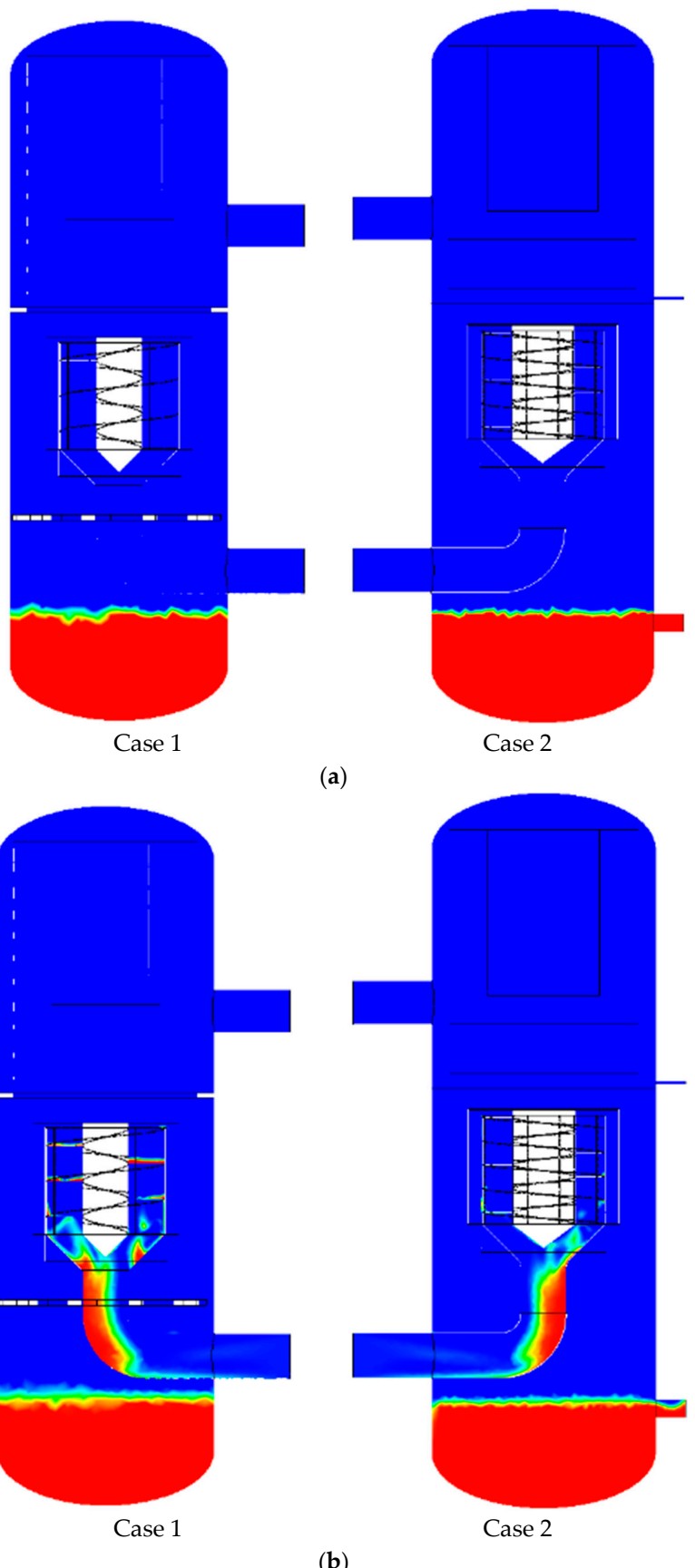

Case 1                      Case 2

(**a**)

Case 1                      Case 2

(**b**)

**Figure 8.** *Cont.*

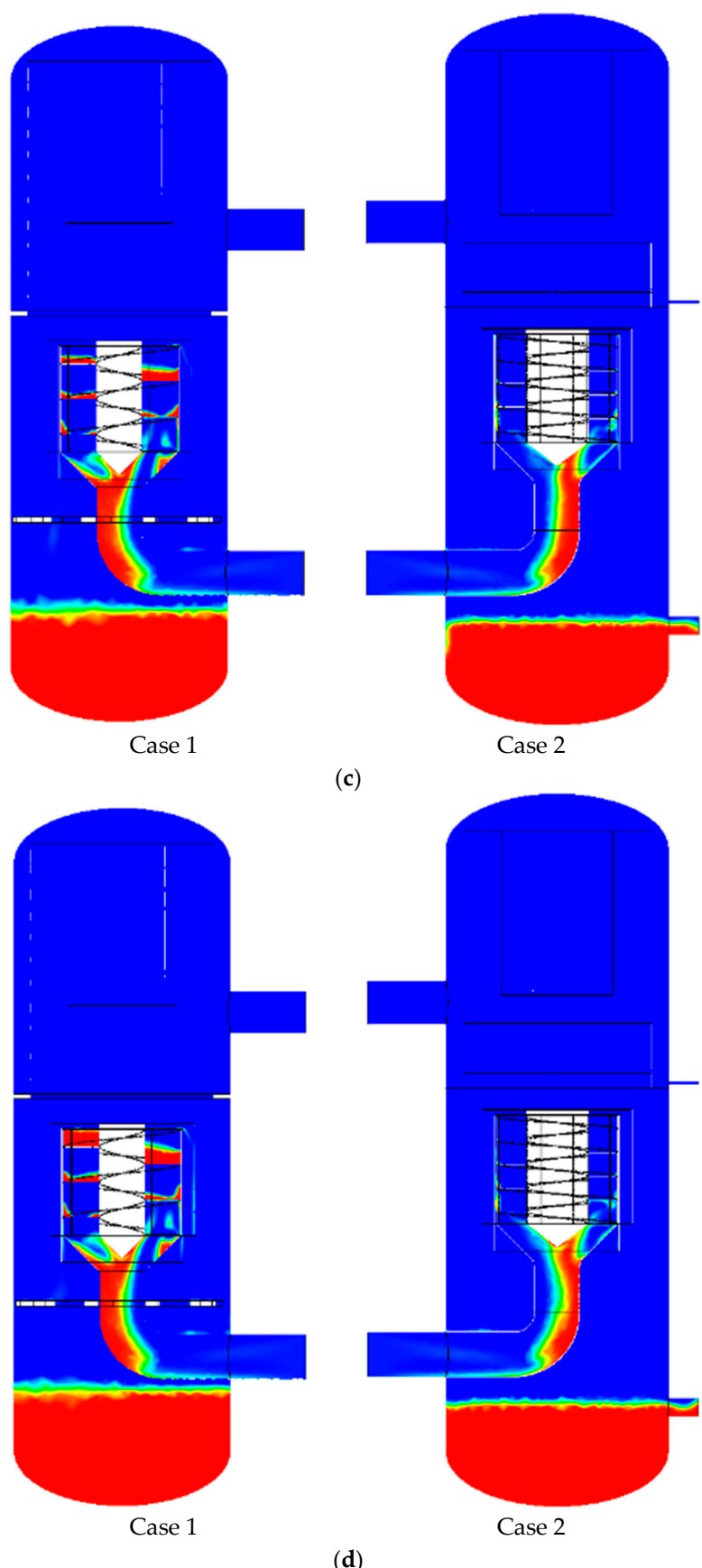

**Figure 8.** *Cont.*

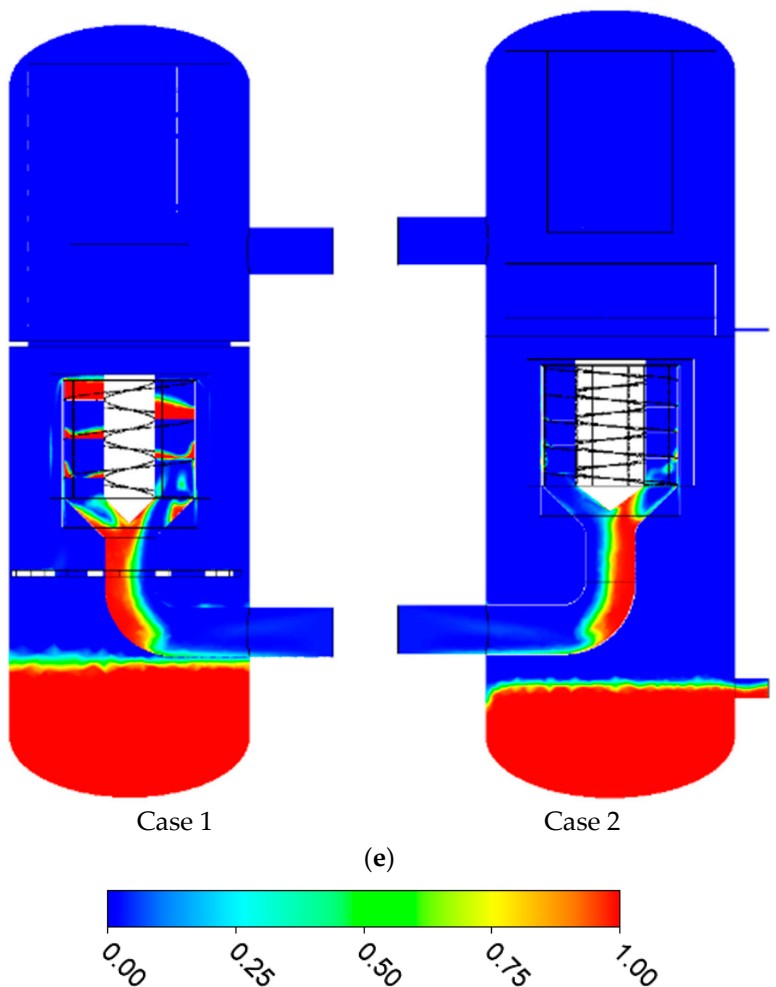

Case 1        Case 2

(**e**)

**Figure 8.** Oil volume fraction: (**a**) 0 s; (**b**) 5 s; (**c**) 10 s; (**d**) 15 s; and(**e**) 20 s.

## 4. Discussion

It is noteworthy that during the experimentation phase, the test installation operated under a variable regime owing to its simultaneous role in supplying compressed air to two other experimental installations developed by INCDT COMOTI. These co-existing experimental setups were also actively engaged in their respective experimentation stages. Consequently, the compression assembly, along with the entire separation system, operated under a distinct operating regime during the experimentation phase, deviating from the originally established regime based on the design stage and the numerical flow simulations conducted beforehand.

Table 2 lists the main parameters of interest that determine the performance of the separation solution

**Table 2.** Main parameters recorded during the experimental phase.

| Parameter | Avg. | Max. | Units |
|---|---|---|---|
| Compressor speed | 699.97 | 2041.39 | rpm. |
| Gas flow | 637.58 | 1845.65 | $Nm^3/h$ |
| Inlet pressure | 5.64 | 13.35 | $bar_g$ |
| Outlet pressure | 5.59 | 13.28 | $bar_g$ |
| Pressure drop | 0.05 | 1.89 | bar |
| Total oil flow | 122.72 | 229.29 | L/min |
| Gas temperature | 58.69 | 81.15 | °C |
| Power consumption | 58.69 | 252.80 | kW |

During this investigation, some observations were made regarding the pressure drop across the system at specific operating points (Figure 9). Upon careful analysis of the recorded data, it was discovered that during the compressor shutdown sequence, when the manual back pressure valve opens to release pressure from the system, the pressure within the separator vessel drops rapidly to zero within a few seconds. This sudden pressure drop results in an increasing gas velocity through the oil separator, leading to a flow rate higher than originally designed. Consequently, this phenomenon sheds light on the impact of transient conditions on the separator's performance, emphasizing the significance of considering such operational scenarios to ensure optimal system functionality.

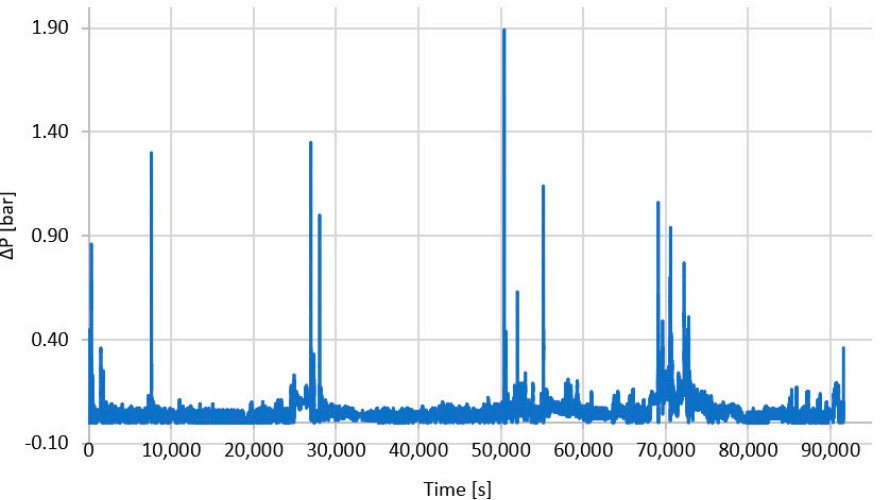

**Figure 9.** The pressure drop in the system during 25 h of operation.

In an industrial setting, the compression assembly typically operates within relatively stable operating parameters that vary slowly over time. Transient regimes, characterized by dynamic changes, primarily occur during the start-up and shut-down stages of the screw compressor assembly. However, a unique scenario resembling the test configuration arises during emergency shutdown events, such as a sudden drop in discharge pressure triggered by actions like pressing the mushroom button or experiencing a station voltage drop.

In such emergencies, the entire gas volume within the assembly, compressed to its working pressure, undergoes rapid evacuation through the basket system due to the PLC-controlled opening of a solenoid valve. This specific case highlights the significance of examining transient behaviors, providing valuable insights into the system's response during critical operational events.

The pressure drop on a separation system over the operating period is an extremely important parameter. The use of a gas compression system working with a high-pressure drop on the gas–oil separation system will result in additional energy consumption.

Analyzing these values, it can be concluded that the experimental setup behaves well in terms of pressure drop at different operating regimes.

It is strongly recommended to adopt a progressive approach when opening the back-pressure holding valve to ensure the optimal functioning of the separation system. A sudden release of compressed gas from a separator vessel can trigger undesired consequences, such as gas–oil separation stage flooding and subsequent oil loss in the exhaust system. Particularly, under specific temperature and pressure conditions, the phenomenon of compressed gas dissolving in oil may arise.

During decompression, the gas molecules mingled with the oil will swiftly discharge, resulting in oil droplet entrainment. To mitigate the risks associated with oil entrainment, alongside the manual or controlled progressive opening of operating elements, it is essential to incorporate calibrated orifice nozzles within the stack exhaust system. These nozzles should be meticulously calculated to enable controlled and gradual re-

lease of the compressed gas, ensuring the preservation of the separator's efficient oil separation performance.

During the experimental phase, the liquid retained in the two collection compartments of the ASF400PN40 coalescer filter was collected twice, at 12 and 25 h of operation (Figure 10), respectively. The drained mixture consisted of a mixture of water, solid suspensions, and oil. The resulting mixture, which still contained a small amount of water, was filtered several times until only oil remained in the measuring vessel. The separation process was relatively difficult. It was expected that only the retained oil would flow into the collection compartments of the coalescer filter, but due to the low working temperature, water vapor from the compressed air condensed and settled on the pipes and bottom of the coalescer filter. The presence of water in the liquid state also caused iron oxide to appear in the compressed air paths, including in the collected mixture.

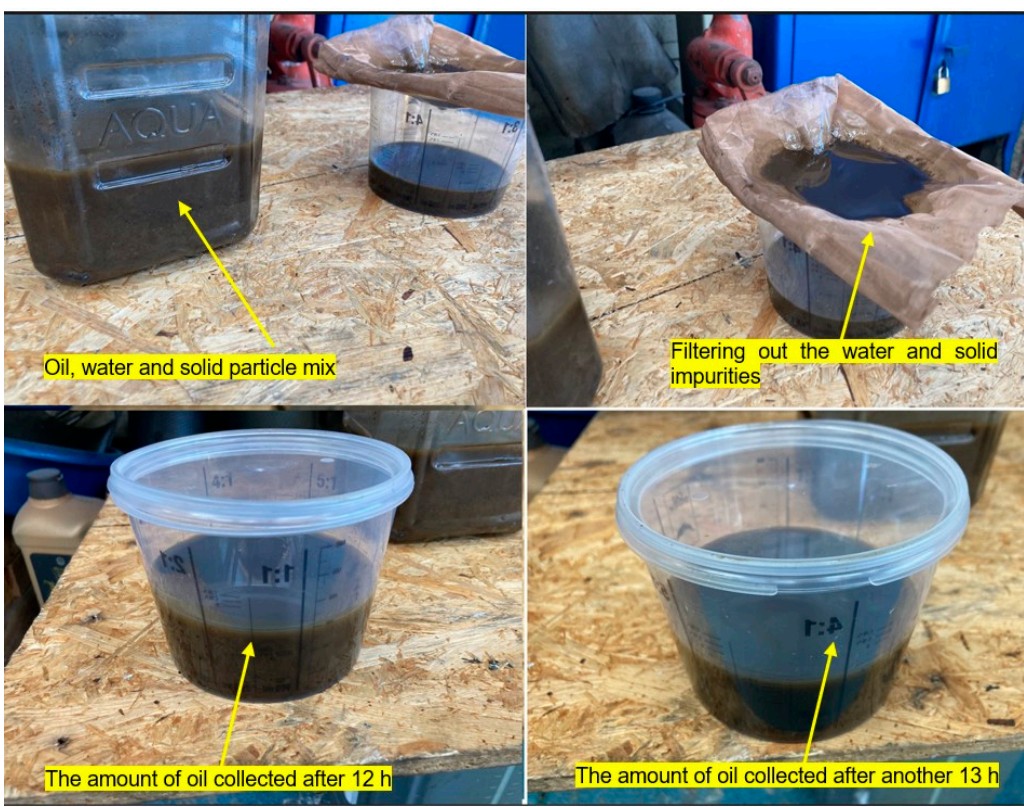

**Figure 10.** Processing of the collected oil samples.

For the calculation of the residual oil quantity at the outlet of the separator vessel, the calculation method previously applied for the conventional separation system according to ISO 8573-1 was used.

$$X = \frac{V \cdot \rho}{q \cdot H \cdot 3600} \cdot 10^6 \qquad (3)$$

where:
  V—The quantity of oil collected (L).
  q—Gas flow at standard conditions (L/s).
  H—Test duration (h).
  ρ—Oil density (kg/m$^3$).
  X—The quantity of residual oil at the separator outlet (mg/Nm$^3$).

After 13 h of operation, the collected amount of oil retained by the coalescer filter was 200 mL. Applying relation (1) resulted in a residual oil content of approx. 20 ppm, expressed in mass ratio.

After 25 h of operation, considering the average compressed air flow rate, the collected oil retained by the coalescer filter was 200 + 250 mL. Applying relation (1) again resulted in a residual oil content of 18 ppm, expressed in mass ratio.

## 5. Conclusions

The experimental validation of the gas–oil separation system effectively corroborated the anticipated performance outcomes from the design and flow simulation stages, affirming its practical applicability and robustness. Withsuccessful implementation and utilization, the system notably enhanced the overall oil retention performance within the closed screw compressor operating system.

The pressure drop serves as a vital performance indicator for the separation system, reflecting its efficiency. Under normal operating conditions, a notable observation emerged, with all separation stages exhibiting an average pressure drop of less than 0.15 bar. However, when the filtered gas flow surpassed thedesigned flow by approximately 50%, a higher pressure drop of 1.89 bar was recorded.

The residual oil content observed at the exit of the system ranged between 19 and 17 parts per million (ppm), signifying an impressive performance that strongly advocates for the approval and adoption of the separation system within industrial applications involving compressed air or combustible gases.

Moreover, coupling the separation system with a coalescing final filter presents a compelling opportunity to attain even more remarkable oil separation outcomes, wherevalues below 5 ppm are achievable. Notably, this enhanced performance remains sustainable for at least 4000 h of operation, all without any noticeable increase in pressure drop and without necessitating the replacement of the coalescing filter elements. This exceptional capability paves the way for prolonged, efficient operation and underscores the system's potential for diverse industrial applications.

The results derived from the experimental tests demonstrated the separation system's performance, signifying a promising milestone in its potential approval for practical implementation within industrial applications involving compressed air or combustible gases. These findings serve as a significant stepping stone toward the system's integration and utilization in real-world industrial scenarios.

The versatility of the separation system allows for its utilization in two distinct domains. Firstly, it can enhance the performance of existing separator vessels, thereby optimizing their efficiency. Secondly, it can be readily integrated into new projects requiring separator vessels, encompassing a diverse spectrum of compression applications. This adaptability ensures its applicability across various scenarios, catering to a wide range of industrial needs.

**Author Contributions:** Conceptualization, S.G.T. and S.V.; methodology, I.M. and R.C.; software, I.M. and S.V.; validation, S.G.T., S.V. and R.C.; formal analysis, I.M.; investigation, I.M.; resources, S.G.T.; data curation, R.C.; writing—original draft preparation, I.M. and R.C.; writing—review and editing, I.M. and S.G.T.; visualization, S.V.; supervision, I.M.; project administration, S.G.T.; funding acquisition, S.G.T. All authors have read and agreed to the published version of this manuscript.

**Funding:** This work was carried out within the "Nucleu" Program EvoTurbo 2023, supported by the Romanian Minister of Research, Innovation and Digitalization, project number PN23.12.05.01. This paper was also supported by the OPTIM Research project, code MySMIS 153735, co-funded by the European Social Fund (FSE) through the Operational Programme Human Capital (POCU) 2014–2020, implemented in 2022–2023.

**Data Availability Statement:** Not applicable.

**Conflicts of Interest:** The authors declare no conflict of interest.

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
