# Peer review of "Experimental Validation of the Numerical Model for Oil–Gas Separation"

_inventions, doi:10.3390/inventions8050125_

Round 1

Reviewer 1 Report

Revise the abstract. It should be more concise and entirely referred to the article; here the first sentences are general and more appropriate for an introduction than for an abstract.

In some cases, the vocabulary should be adjusted. For instance, the expression "Experimental experiments" must be modified.

Section 2 might be enriched by adding the equations which are currently only mentioned at the beginning of the same section.

Table 1: the temperature is given in °C, but in the other parameters where it appears, as Specific heat capacity and other, it is in K. Please adopt only one unit for temperature.

Section 4 and Section 5 have the same name, probably the related paragraphs can be put together.

The writing only needs of minor revision in my opinion.

Author Response

1. Revise the abstract. It should be more concise and entirely referred to the article; here the first sentences are general and more appropriate for an introduction than for an abstract.

The Abstract section was completely rewritten, with a sentence for context and background, motivation, hypothesis, methods, results, and conclusions.

2. In some cases, the vocabulary should be adjusted. For instance, the expression "Experimental experiments" must be modified.

We corrected that error.

3. Section 2 might be enriched by adding the equations which are currently only mentioned at the beginning of the same section.

Equations for the numerical model were added.

4. Table 1: the temperature is given in °C, but in the other parameters where it appears, as Specific heat capacity and other, it is in K. Please adopt only one unit for temperature.

We adopted the unit Kelvin for the temperature.

5. Section 4 and Section 5 have the same name, probably the related paragraphs can be put together.

According to the requirements of the Journal, we redid the sections of the paper and now sections 4 and 5 along with 2 and 3 are placed in Chapter 2. Materials and Methods.

Reviewer 2 Report

In gas separators, a number of designs of which are considered in this article, a mixture with a relatively small content of liquid phase is processed. With all the variety of technological schemes, the most important parameter of separators is the separation efficiency, which means the maximum yield of liquid and gas phases at the minimum residue of unextracted product. The paper presents the results of hydrodynamic modeling using CFD (Computational Fluid Dynamics) with the use of Volume of Fluid and Discrete Phase Model methods for a complex separator design including a cyclone with centrifugal-gravity separation mode and filtration through porous medium. The use of these methods allowed to obtain a visual image of the trajectories of fluid particles in the field of gas flow at different modes and places of mixture feeding.  An important result of the work is the experimental verification of the modeling results, which showed a sufficiently high separation efficiency. 

Some inaccuracies in the text are summarized below:

1. Almost everywhere there is no space before the square brackets of literature references.

2. Table 1 incorrectly uses "," and "."

3. In Table 1, the dimension of density Kg/m3 should be kg/m3.

4. In the caption to Figure 5... e) 20, should be e) 20 s.

6. line 186: range,allowing needs a space.

7. Figure 5. fraction::

8. The pictures in Figure 9 "...after 12 h" and "...after another 13 h" appear to be the same.

9. line 309 - "residual oil content 20.554", the accuracy of the determination seems overstated.

10. line 323 "thedesigned", could it be "the designed" ?

11. In the reference list, not all sources have doi.

Author Response

  1. Almost everywhere there is no space before the square brackets of literature references.

 The request has been fulfilled.

  1. Table 1 incorrectly uses "," and "."

We used "," for the decimal separator.

  1. In Table 1, the dimension of density Kg/m3 should be kg/m3.

The type-o has been corrected.

  1. In the caption to Figure 5... e) 20, should be e) 20 s.

We put the ''s" in Figure 5.

  1. line 186: range,allowing needs a space.

Between "," and the following word, a space was utilized.

  1. Figure 5. fraction::

One set of ":" was deleted.

  1. The pictures in Figure 9 "...after 12 h" and "...after another 13 h" appear to be the same.

The figure was updated.

  1. line 309 - "residual oil content 20.554", the accuracy of the determination seems overstated.

The value for the residual oil content was rounded to 20 ppm.

  1. line 323 "thedesigned", could it be "the designed" ?

Between these two words, a space was utilized.

  1. In the reference list, not all sources have doi.

We try to put the DOI for all the references.

Reviewer 3 Report

Many previous works have been dedicated to the modelling and simulation of oil-gas separators, and even three-phase separators, including comprehensive reviews. Many of them are missing in the literature review, and they need to be included. The manuscript contribution must be highlighted.   

The manuscript presents valuable data on the discharge of lubricating oil. However, it is out of the manuscript's main subject.

The introduction and motivation must be revised to clarify the manuscript's focus. The authors mention flow velocity drops, baffles, and coalescence, whereas the oil-gas separation will be the focus, as it is indicated in the abstract.

The title in Figure 9 is wrong (it is the same as Figure 8)

Author Response

  1. Many previous works have been dedicated to the modelling and simulation of oil-gas separators, and even three-phase separators, including comprehensive reviews. Many of them are missing in the literature review, and they need to be included. The manuscript contribution must be highlighted.

The state of the art was improved by using more current references. In addition, our work was emphasized by the inclusion of one paragraph at the very end of the Introduction chapter.

  1. The manuscript presents valuable data on the discharge of lubricating oil. However, it is out of the manuscript's main subject.

The purpose of the paper was revised to coincide with the substance of the article.

  1. The introduction and motivation must be revised to clarify the manuscript's focus. The authors mention flow velocity drops, baffles, and coalescence, whereas the oil-gas separation will be the focus, as it is indicated in the abstract.

The Abstract section was completely rewritten, with a sentence for context and background, motivation, hypothesis, methods, results, and conclusions.

  1. The title in Figure 9 is wrong (it is the same as Figure 8)

The mistake has been repaired.

Round 2

Reviewer 1 Report

The authors revised the article according to my suggestions; therefore I would recommend to consider this article for publication, in its present form.

English language is suitable for publication.

Author Response

Thank you for your suggestions. We feel that they assisted us in improving the quality of our paper. We double-checked the document and discovered a few mistakes, which we corrected.

Reviewer 3 Report

The authors have revised the manuscript appropriately. The issue is only on the absence of the data from previous works describing the same subject, as it was highlighted before. This absence restricts the discussion of results.

Author Response

Thank you for your time and advice. We included a few references to studies conducted by other researchers on the same oil-gas separation phenomena.